# Brain Activities Show There Is Nothing Like a Real Friend in Contrast to Influencers and Other Celebrities

**DOI:** 10.3390/brainsci13050831

**Published:** 2023-05-21

**Authors:** Peter Walla, Dimitrios Külzer, Annika Leeb, Lena Moidl, Stefan Kalt

**Affiliations:** 1Freud CanBeLab, Faculty of Psychology, Sigmund Freud University, Sigmund Freud Platz 1, 1020 Vienna, Austria; 2Faculty of Medicine, Sigmund Freud University, Sigmund Freud Platz 3, 1020 Vienna, Austria; 3School of Psychology, Newcastle University, University Drive, Callaghan, NSW 2308, Australia

**Keywords:** electroencephalography, event-related potential (ERP), social media, influencer, celebrity, friend, fake friendship, non-conscious processing

## Abstract

Especially for young people, influencers and other celebrities followed on social media evoke affective closeness that in their young minds seems real even though it is fake. Such fake friendships are potentially problematic because of their felt reality on the consumer side while lacking any inversely felt true closeness. The question arises if the unilateral friendship of a social media user is equal or at least similar to real reciprocal friendship. Instead of asking social media users for explicit responses (conscious deliberation), the present exploratory study aimed to answer this question with the help of brain imaging technology. Thirty young participants were first invited to provide individual lists including (i) twenty names of their most followed and loved influencers or other celebrities (fake friend names), (ii) twenty names of loved real friends and relatives (real friend names) as well as (iii) twenty names they do not feel any closeness to (no friend names). They then came to the Freud CanBeLab (Cognitive and Affective Neuroscience and Behavior Lab) where they were shown their selected names in a random sequence (two rounds), while their brain activities were recorded via electroencephalography (EEG) and later calculated into event-related potentials (ERPs). We found short (ca. 100 ms) left frontal brain activity starting at around 250 ms post-stimulus to process real friend and no friend names similarly, while both ERPs differed from those elicited by fake friend names. This is followed by a longer effect (ca. 400 ms), where left and right frontal and temporoparietal ERPs also differed between fake and real friend names, but at this later processing stage, no friend names elicited similar brain activities to fake friend names in those regions. In general, real friend names elicited the most negative going brain potentials (interpreted as highest brain activation levels). These exploratory findings represent objective empirical evidence that the human brain clearly distinguishes between influencers or other celebrities and close people out of real life even though subjective feelings of closeness and trust can be similar. In summary, brain imaging shows there is nothing like a real friend. The findings of this study might be seen as a starting point for future studies using ERPs to investigate social media impact and topics such as fake friendship.

## 1. Introduction

In 1995, Classmates.com was created as the first social networking platform followed by Sixdegrees.com in 1997. Whereas those early platforms were made for connecting with other people by collecting personal profiles, it was in the early 2000s that such platforms began to include instant messaging, which had already been invented separately before. A few years later, both services were available through mobile devices, and today, staying connected and instant messaging are among the most widely used smartphone applications around the world. Those highly desired services together with other internet-based tools are known as social media. The initial idea, as the term social media suggests, was to connect people, to communicate online and live with others regardless of how far away, to share information as wide as around the entire globe in an instant, and to ease international collaboration. However, despite those truly positive features, negative consequences were also soon recognized [1]. Those are at least threefold!

First, an article published in 2021 reviewed no less than 25 distinct theories related to social media addiction [2], which refers to maladaptive social media use leading to behavioral addiction symptoms [3]. However, addiction is only one perhaps unexpected and unwanted consequence of social media use per se mainly leading to a lack of time for more healthy and useful activities [4]. There are also content-related problems related to cyber-mobbing [5], bullying and stalking among various other forms of genuine harm especially to young generations. A further interesting content-related issue of social media, though, does not really demonstrate obvious and direct problems, and that is the concept of fake friends (or virtual friends) [6,7]. Fake online friendship is induced through influencers, bloggers, YouTubers and others that are not interested in reciprocal friendships, while clearly trying to create closeness and trust in their followers.

Although the definition of friendship has been found difficult by a number of sociologists in the past, e.g., [8,9], there clearly are crucial differences compared to what influencers, bloggers and YouTubers mean to us. For instance, Blau [10] investigated the topic of friendship by asking questions to people of varying generations and analyzing their responses. The result has been an understanding of friendship being restricted to a very limited set of close friends. Allan [11] summed up three important points for a friendship understood as a personal relationship. Point 1 is that it must be a relationship between individuals, point 2 is that it must be a private relationship and point 3 is that it must involve the person as the person they really are.

From a neuroethological perspective, friendship is defined through the quality of interactions between individuals. Two humans are friends when they engage in a bidirectional and affiliative way with a higher frequency than with nonfriends. Respective interactions should be nonreproductive and consistent over time [12,13]. According to Dunbar [14], friends share their lives in a way different from just casually meeting strangers including emotional bonds and regular contact.

Whereas a group relationship has been understood as between mutually substitutable people, friendship is a relationship with someone not replaceable [15]. Crucially, a very important aspect of friendship as mentioned in various reports is its non-exploitive nature [e.g., 11]. Joy has to be the driving motive, not some instrumental reason [11]. This latter point might be most important in the context of influencers, bloggers and YouTubers.

While a young social media consumer supports influencers, is loyal and feels close to them, the influencers on the other hand are only interested in influencing, hence the label influencer. They want something from their social media consumers that is clearly meant to lead to financial gain; it is not true friendship they are looking for [16,17].

Found on platforms such as Facebook, Instagram, YouTube, Twitter and Snapchat, influencers are expanding their network to make money by presenting sponsored products [18]. Many influencers became widely known celebrities that especially young social media consumers follow every single day. Many if not most social media consumers have strong feelings of genuine friendship and trust [19]. What used to be band singers or film stars that were perceived as celebrities are now influencers, especially in the young generation. Influencers match the status of a celebrity and even more because an influencer’s intention is to make money by creating a fake friendship (via entertaining the follower) that is meant to elicit trust in a follower finally persuading him to buy products. Of course, such fake friendship is only one-way and thus potentially problematic [6].

Even though the topic of friendship has been researched a lot in the past (see above), what has been neglected so far is the use of brain imaging technology, especially in the context of the fresh and acute problem of fake friendship of influencers and other celebrities. This problem was also raised in a recent review report by Giumetti and Kuwalski [5] when they stated that most of the existing research is self-reported, which introduces problems of social desirability bias [20] in addition to misleading data because affective brain responses (preference is strongly affective) guiding human behavior are not easy to verbalize [21], a concept that has been labeled “cognitive pollution” [22,23]. More objective experimental research is needed, and the present study was meant to provide empirical insight into this very acute topic by conducting a brain imaging study (electroencephalography) analyzing brain activities elicited by individually selected names of followed and loved influencers and other celebrities while comparing those to brain activities elicited by names of real-life loved friends and relatives as well as elicited by known names without any affiliation. For this purpose, thirty young participants were asked to provide individual lists including names for all three categories before they were invited for a lab visit (Freud CanBeLab). While they were then exposed to their individual name lists, their brain activities were recorded and later calculated into ERPs, which were statistically analyzed. This study is not hypothesis-driven; it is exploratory, aiming at presenting ERP data that distinguish (in the absence of conscious deliberation) between the abovementioned three name categories by using a neurophysiological method providing the highest temporal resolution.

## 2. Materials and Methods

### 2.1. Participants

In total, 30 participants were invited to the Freud CanBeLab (Freud Cognitive and Affective Neuroscience and Behavior Lab) (https://psychologie.sfu.ac.at/de/fakultaet/institute/freud-canbelab/; accessed on 10 May 2023) after they provided the required name lists (see Section 2.2. Stimuli). Two participants had to be excluded due to measurement artifacts. The remaining group of 28 participants contained 18 males and 10 females. Their mean age was 21.93 years (SD = 1.61). They were all right-handed and had no neuropathological history. As part of the demographics survey, they were also asked how many hours per day they consume social media. Sixteen participants reported consuming social media for 1–2 h per day, seven reported 2–3 h per day consumption and five reported consuming social media for more than 3 h. Those results are not further analyzed; they are only meant to demonstrate that all participants are social media users. They signed a consent form and were informed that they could withdraw from the study at any time during the experiment without any consequences. The study received ethical approval from the Sigmund Freud University Ethics Committee (approval code: XCQCDPNWBPBCOM90080).

### 2.2. Stimuli

Prior to their lab visit, all participants were asked to provide twenty names of their most loved and followed influencers and celebrities (fake friend names), twenty names of loved friends and relatives (real friend names) and twenty people’s names that do not mean anything to them (no friend names). Those individual lists containing sixty names were then used to program experiments for each participant separately. Stimulus presentation was administered and controlled by the free software PsychoPy2 for Windows [24]. The programmed software scripts were designed to also send triggers to the electroencephalography (EEG) system in order to provide condition coding for later EEG and ERP data analysis.

### 2.3. Procedure

After arrival at the lab, the participants were introduced to the purpose of the study. They were given the informed consent form to sign if they agreed to participate. The actiCAP with 64 electrodes embedded (from Brain Products) was applied and connected to the amplifier (see further details below). Before the recordings started, the participants were instructed to sit still and blink with their eyes only when they saw a fixation cross, but to avoid blinking during name presentations. Each name was presented twice (in a random order; 40 presentations per condition) for 300 ms in white letters on a black background on a computer monitor placed on a table in front of the participants, who sat on a comfortable chair. This was followed by a blank black screen for 1 s and a white fixation cross on a black background for 1 s with a final blank black screen for again 1 s. The eye-to-screen distance was about 1 m, and all visual stimuli were presented so as to stimulate foveal receptors only (no peripheral field stimulation). The participants were instructed to indicate via a button press whether they saw a real friend name, a fake friend name or a no friend name.

### 2.4. Electroencephalography (EEG)

For recording brain potential changes, a 64-channel actiCHamp Plus System from Brain Products with active electrodes embedded in an actiCAP connected to an amplifier was used. The amplifier was operated by a powerful lithium-ion battery pack. Brain potentials were sampled with a rate of 1 kHz (filtered: DC to 100 Hz). Impedance was kept equal to or below 10 kΩ. Cz was used as the reference electrode and a midfrontal position on the forehead was used for the ground electrode. Offline, all EEG data were down-sampled to 250 Hz and a bandpass filter from 0.1 to 30 Hz was applied in preparation for following EEG data processing.

### 2.5. Analyses

EEG data processing was carried out with an updated version (6.4.9) of the initial EEGDISPLAY 6.1.5 software, which was developed by Fulham [25]. Epochs from 100 ms before stimulus onset (baseline) until 1 s after stimulus onset were generated. All epochs contaminated by visible artifacts were manually selected and excluded, and those with the electrooculogram (EOG) amplitude exceeding ± 75 mV were automatically excluded. Event-related potentials (ERPs) were calculated for each of the three conditions of interest for each participant. Finally, grand averages were calculated for each condition of interest across all 28 participants including all 64 electrodes to be displayed in the results section. Out of a maximum of 40 possible trials per name condition, the mean number of trials included in the fake friend name ERP per subject was 28.21 (SD = 8.37), in the real friend name ERP, it was 30.36 (SD = 7.26) and in the no friend name ERP, it was 30.5 (SD = 7.02). A visual inspection of the overlaid ERPs at all 64 electrodes (see Figure 1) resulted in the decision to focus the following statistical analysis on four selected electrode locations (FT9 and FT10 (left and right frontotemporal) and TP9 and TP10 (left and right temporoparietal)) that showed the most prominent ERP differences. Respective ERPs from those four locations are shown in Figure 2. For statistical analysis, EEG data from all participants were further down-sampled resulting in data points averaged across 20 ms time windows spanning from 200 ms to 820 ms after stimulus onset. With those data points (amplitude values), first, an ANOVA (analysis of variance; repeated measures; Greenhouse–Geisser corrected) was calculated for every single 20 ms time window following a 3 × 2 × 2 experimental design. The first factor *social name* had 3 levels, “fake friend names”, “real friend names” and “no friend names”. The second factor *hemisphere* had 2 levels, “left hemisphere” and “right hemisphere”. The third factor *electrode* had 2 levels, “frontal” and “parietal”. Due to multiple comparisons, the Benjamini–Hochberg correction method [25] was applied. Finally, paired sample *t*-tests were calculated to compare each possible pair of name conditions for each of the four electrodes for representative time windows that showed significant ANOVA effects. Because of multiple comparisons, we again applied the Benjamini–Hochberg method [26]. Finally, topographical maps were created to show color-coded brain amplitudes over the whole scalp for each name condition separately for selected representative time points showing significant effects (Figure 3).

## 3. Results

### 3.1. Event-Related Potentials (ERPs) and Topographical Maps

As already presented in the method section, ERPs calculated for all three social name conditions and finally overlapped show a wide spectrum of brain amplitude differences (see Figure 1). However, the most dominant effects are seen in the left and right frontal and left and right temporoparietal regions (most dominantly at FT9, FT10, TP9 and TP10). As already mentioned in the method section, data collected from those four electrode locations were selected for further statistical analysis. Figure 2 shows magnified ERPs from those four locations.

### 3.2. Analytical statistics

The analysis of variance (ANOVA) including the factors *social name*, *electrode* and *hemisphere* resulted in two separate time periods where main *social name* effects occurred (early and late effect). Even though only one data point representing the earlier time window from 300 ms to 320 ms survived the Benjamini–Hochberg correction, the time window before and after show strong trends towards significant main effects of the *social name* condition as well (see Table 1). Slightly later, a longer sequence of time windows shows significant *social name* condition effects spanning from 400 ms to 700 ms after stimulus onset. Table 1 shows all ANOVA results.

In the following, *t*-tests were conducted to compare each possible pair of name conditions for each of the four electrodes and for both the early time window and one of the later time windows representing the later effect, where ANOVA results showed significant *social name* main effects. For electrode location FT9, for the 300 ms to 320 ms time window, fake friend names elicited a significantly less negative amplitude compared to real friend names and also compared to no friend names (this difference even survived Benjamini–Hochberg correction) (see Table 2). No difference was found between real friend names and no friend names. No such differences were found at this time for any other electrode location with the only exception of a significant difference between fake and real friend names at electrode location TP9, where real friend names elicited more negative going potentials. For the 500 ms to 520 ms time window and for all four electrode locations, fake friend names also elicited a significantly different amplitude compared to real friend names, but no longer compared to no friend names, which elicited similar brain potentials to fake friend names. Thus, at this time point, real friend names elicited brain potentials that strongly differed from both other conditions (see Table 2).

At electrode locations FT10 and TP10 for the early time window, no differences occurred between any name conditions. This is seen as strong evidence that the early effect (300 to 320 ms) described above is indeed left-dominant. However, all electrode locations show the same pattern of ERP differences regarding the later effect. This means that at all four locations, for the later time window, real friend names elicited significantly more negative brain amplitudes compared to both other name conditions. It also means that for the later time window, fake friend names were processed by the brain very similarly to no friend names, which is interpreted as empirical, neurophysiological evidence that the brain is not deceived by the actual fakeness of influencers and other celebrities.

## 4. Discussion

The findings of this study might have implications regarding the influences of social media. We ensured that all our study participants can be classified as social media users by asking them how many hours they consume social media per day. However, we could not further analyze the respective data because of too few participants. Future studies comparing high versus low users should follow. However, the findings of this study are largely interpreted as depending on social media impact.

Friendship is a widely investigated research topic [27]. In rare cases, it has been investigated by utilizing brain imaging technology, and one of the main findings was that similarities in neural responses to controlled stimulation were able to predict friendship [28]. The morphology and thickness of neural structures have been found to correlate with friendship quality [29]. An fMRI study revealed that the activation level of the nucleus accumbens correlated with personal reward as well as vicarious rewards for both parents, but not with reward for a stranger [30]. In contrast to friendship as such, the concept of fake friendship, as it occurs in the frame of social media, has not been researched well yet. In a very recent systematic review of social media use and its negative aspects, the authors mention in their abstract that “most research has relied on self-report and cross-sectional examinations of these constructs” [31]. This point has already been raised in the introduction. In contrast to this widespread approach, the present empirical investigation used brain imaging technology to contribute to a better understanding of one certain aspect of social media impact, namely, fake friendship (i.e., unilateral closeness) of influencers and other celebrities. The rather exploratory (to our knowledge, there is no ERP study on this topic published yet) question of interest was if the brain processes names of followed and loved influencers and other celebrities (fake friend names), names of actual loved friends and relatives (real friend names) and names that do not elicit any feeling of closeness (no friend names) differently. EEG was utilized to measure the brain’s responses to visual presentations of those names (presented in random order), and ERPs were calculated and compared. Overall, this exploratory investigation revealed that the brain indeed processes those name categories differently, even though the felt familiarity aspect of fake and real friends is similar. Given that the vast majority if not all prior investigations on this topic were conducted on the basis of self-reporting any comparison to existing literature is difficult, which gives this study its exploratory nature, potentially leading to new hypotheses. To the best of our knowledge, the most similar study to our investigation compared ERPs related to one’s own name, a famous name, the name of a close other person and an unknown name [32]. The authors found that the name of a close other person elicited similar brain potentials to one’s own name, while both categories differed from a famous name and an unknown name. Assuming that their category “famous name” is similar to our “fake friend name” condition and that their category “close other name” resembles our “real friend name” condition, their results can be linked to our findings. They also found differences between “famous name” and “close other name”; however, their results are difficult to interpret, because they presented a single name repeatedly for each condition to allow for the generation of ERPs. This introduces unwanted repetition effects that are potentially different for the different name categories. Nevertheless, our findings are similar and further investigations including ERPs to describe social media impacts or similar topics seem meaningful.

The first finding from the present study was a short (ca. 100 ms duration) effect in the left frontotemporal region starting at around 250 ms post-stimulus onset. It shows similar brain activation levels for real friend names and no friend names, while both differ from brain activity elicited by fake friend names. Real friend names and no friend names elicited significantly more negative going brain potentials compared to fake friend names, which is interpreted as higher cortical brain activation in the respective region. The second finding is a slightly longer effect (ca. 250 ms) starting at around 300 ms post-stimulus onset. While the first effect was quite focal (left frontotemporal), the second effect is more widespread and was found in the left and right frontotemporal as well as the left and right temporoparietal regions. It shows that in those regions, real friend names elicited the by far most negative going brain potentials compared to both other conditions (fake friend names and no friend names). Although it is not easy to interpret this later effect, it, first of all, shows that real friend names are processed significantly different to fake friend names, which the brain seems to process similarly to strangers’ names (i.e., no friend names). However, we would like to mention that the condition “real friends”, which includes actual loved friends as well as loved relatives, might be understood as representing close loved people in general and not particularly friends. It might seem unlikely, especially for an older generation, that one can have 20 close friends. Consequently, the condition “real friends” might be seen as a limitation in terms of terminology. In addition, due to the innovative nature of this study in terms of using ERPs to investigate social media’s impact, or, more accurately, brain processes related to people’s names varying in real versus fake closeness (or in other words reciprocal versus unilateral closeness), our findings should indeed (as mentioned above) more be seen as leading to new hypotheses rather than answering questions.

The use of EEG, in particular, the analysis of ERPs, is very novel in this field. EEG is known for its excellent temporal resolution (e.g., [33]), which is mirrored in the capacity to capture even short brain functions such as the two findings from the present study, but it is also known for its poor spatial abilities. Consequently, it is difficult to interpret both findings regarding their spatial features in terms of actual neural correlates. In addition, neurophysiological effects occurring at certain electrode locations are not necessarily happening right underneath those locations. However, it is reasonable to link the left frontotemporal region, which has been described as being recruited in the frame of verbal fluency tasks (categorical and letter fluency) [34] with the first finding that fake friend names were processed differently from real friend and no friend names. Both versions of verbal fluency tasks measure memory [35], language [36] and executive function [37]. Especially, language processing (mainly semantic aspects) functions might be relevant, which is also due to the well-known Broca area located left fronto-temporally [38]. The Broca area has not only been found involved in language processing but also music [39]. It seems crucial for syntax processing in language as well as music, which is consistent with the notion that Broca’s area processes syntax in a rather general way. In other words, one could argue that it also provides rather holistic processing. In the current linguistic context, syntax is understood as referring to the arrangement of words in sentences leading to a rather holistic meaning of a whole sentence. In the present study, where only personal names were visually presented, it could still mean that holistic processing of real friend and no friend names differed from fake friend names at or around Broca’s area.

As mentioned above, the vast majority of studies on the impact of social media on young humans are survey-based. As outlined in the introduction, the consumption of social media has been reported as leading to negative consequences. To mention another study, Simsek et al. [40] found that high school and university students in Turkey have moderate levels of social media addiction. While such studies are survey-based investigations, the present study is, to our knowledge, the first empirical report that applied a method that does not require any explicit verbal response (i.e., ERPs) in this field of interest.

The rise of smartphones providing mobile internet access changed a lot. Social media consumption suddenly became available out of a trouser pocket, and fake friendship became a chance to occur. In 2011, only a few years after the first smartphones appeared, O’Keefe and Clarke-Pearson [1] published a report on the impact of social media on children, adolescents and families. Besides a couple of benefits such as enhanced learning opportunities, the authors also raise awareness in their article about various risks. In addition to those mentioned already in the introduction, among those are cyberbullying [41], online harassment, sexting (sending and receiving sexually explicit messages) and Facebook depression [42]. However, the results of our study give hope that our brain is, despite all those negative consequences, at least dealing differently with fake friends such as influencers and other celebrities compared to real friends offering reciprocal friendship. Without neglecting the obvious drawbacks of social media, it seems as if our brain is not easy to trick, at least not with respect to its non-conscious processing. Thus, even though conscious cognitive closeness to influencers and other celebrities seems real, the brain itself clearly knows the difference. Hence, there is nothing like a real friend.

Regarding limitations, we would like to summarize that first of all our study is exploratory, which means that we did not state any hypotheses. Future studies should arise that are more hypotheses-driven. As already mentioned, the concept of friendship is ambiguous, which means that for some, it seems unlikely to have twenty loved people in real life as well as twenty influencers or other celebrities that one follows on a regular basis. Consequently, some aspects of our interpretation need to be handled with caution, especially when it comes to talking about “fake friendship” versus “real friendship”. Importantly, it has to be emphasized again that anatomical interpretations are potentially misleading, which lies in the nature of ERP data. Finally, future studies could implement other computational methods such as permutation and general linear modeling [43,44,45]. In addition, source localization analysis as well as other brain imaging tools such as functional magnet resonance imaging (fMRI) could be applied.

## 5. Conclusions

Importantly, the findings of this study should not be understood as supporting the consumption of influencer-based social media in young adults. Instead, it can be seen as empirical evidence that the human brain, at least on a non-conscious level, is capable of distinguishing between a followed, loved celebrity (unilateral friendship) and a loved person out of real life (reciprocal friendship). Potentially, this ERP study can be seen as a starting point for future investigations completing knowledge about social media impacts gained via survey investigations with objective technology, in particular with event-related potentials (ERPs). Clearly, relying solely on explicit responses, which is the main clinical approach so far, might be misleading, because the brain knows more than it admits to consciousness. From a socio-behavioral perspective, we want to emphasize that, even though the findings from this study might take away some of the expected negative aspects of social media use, it has to be emphasized that time spent on social media is still a major problem [4].

## Figures and Tables

**Figure 1 brainsci-13-00831-f001:**
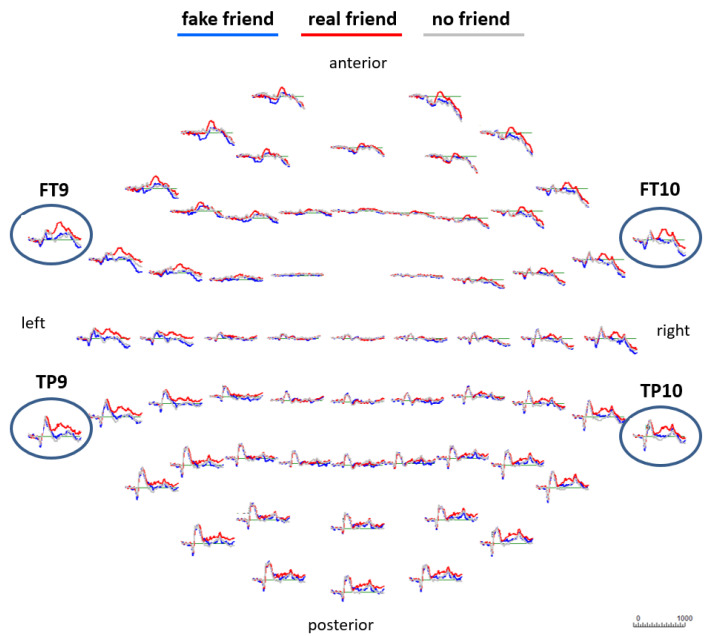
Event-related potentials (ERPs) calculated for every single electrode location. Most dominant effects occurred at frontotemporal and temporoparietal brain regions. The selected electrodes are marked with blue ellipses. See Figure 2 that shows ERPs from those four electrode locations magnified.

**Figure 2 brainsci-13-00831-f002:**
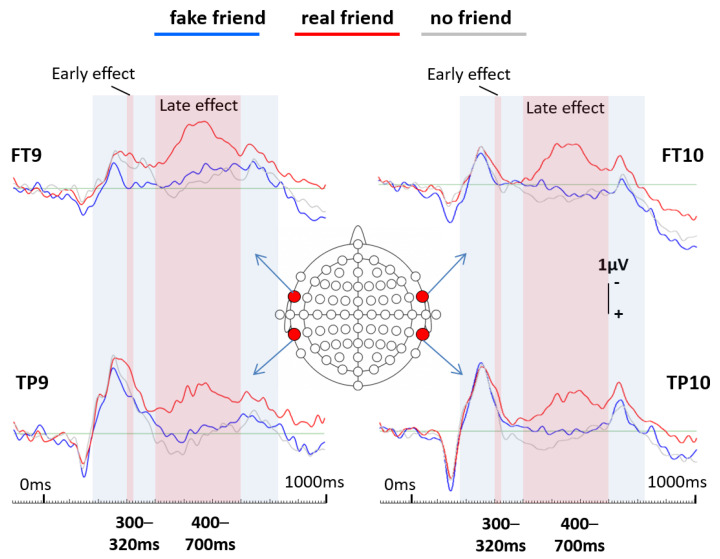
Overlaid event-related potentials (ERPs) for each of the three social name conditions at the four selected electrode locations FT9 and TP9 (left hemisphere) and FT10 and TP10 (right hemisphere). Marked in light red color are the time windows showing significant *social name* effects. The peak of the early left frontotemporal effect ranges from 300 ms to 320 ms and the late overall effect ranges from 400 ms to 700 ms.

**Figure 3 brainsci-13-00831-f003:**
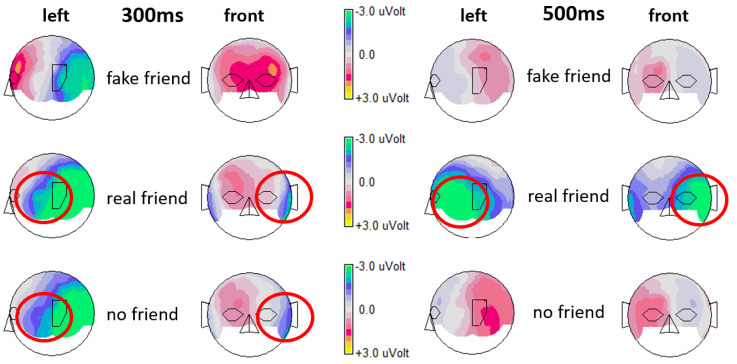
Topographical maps including data from all electrode locations created for all three name conditions, fake friend names, real friend names and no friend names. Maps on the left were created for a time point representing the early effect (300 ms). Maps on the right were created for a time point representing the late effect (500 ms). Note that the left maps show similar topographies for real friend names and no friend names (including a left frontotemporal region; marked by red circles), whereas the right maps show that real friend names elicited most negative brain amplitudes compared to both other conditions slightly later (marked by red circles).

**Table 1 brainsci-13-00831-t001:** ANOVA results. Shown are *p*-values related to *social name* main effects for each 20 ms time window separately (from 200 ms to 820 ms after stimulus onset). Due to multiple comparisons, Benjamini–Hochberg (B.H.) adjusted *p*-values are also displayed (right column) (significant *p*-values are bold).

Time Window	*Social Name* Main Effect			
	*p*-Value	F	df	B.H. Adjusted
200–220	0.357	0.992	0.035	0.395
220–240	0.954	0.022	0.001	0.954
240–260	0.674	0.379	0.014	0.720
260–280	0.291	1.259	0.045	0.385
280–300	0.034	3.677	0.120	0.062
300–320	0.020	4.575	0.145	**0.045**
320–340	0.053	3.171	0.105	0.091
340–360	0.326	1.144	0.041	0.387
360–380	0.756	0.251	0.009	0.781
380–400	0.191	1.736	0.060	0.282
400–420	0.022	4.192	0.134	**0.045**
420–440	0.009	5.354	0.165	**0.023**
440–460	0.002	7.446	0.216	**0.006**
460–480	<0.001	12.670	0.319	**0.004**
480–500	<0.001	17.461	0.393	**0.004**
500–520	<0.001	18.898	0.412	**0.004**
520–540	<0.001	11.917	0.306	**0.004**
540–560	<0.001	14.998	0.357	**0.004**
560–580	<0.001	19.040	0.414	**0.004**
580–600	<0.001	16.781	0.383	**0.004**
600–620	<0.001	10.971	0.289	**0.004**
620–640	0.002	7.110	0.208	**0.006**
640–660	0.005	6.184	0.186	**0.014**
660–680	0.027	4.075	0.131	0.052
680–700	0.021	4.308	0.138	**0.045**
700–720	0.056	3.154	0.105	0.091
720–740	0.059	3.112	0.103	0.091
740–760	0.337	1.093	0.039	0.387
760–780	0.298	1.224	0.043	0.385
780–800	0.313	1.170	0.042	0.387
800–820	0.292	1.246	0.044	0.385

**Table 2 brainsci-13-00831-t002:** *t*-Test results for all four electrodes for the early time window (300 ms to 320 ms) that showed a significant main *social name* effect and for one representative later time window, where a series of consecutive time windows showed significant main *social name* effects (500 ms to 520 ms). Because of multiple comparisons, Benjamini–Hochberg correction was applied (significant *p*-values are bold).

Electrode and Time	*t*-Test Pairs	T	df	*p*-Value	B.H. Adjusted
FT9–300–320 ms	Fake-real	2.949	27	0.007	**0.019**
	fake-no	2.626	27	0.014	**0.034**
	real-no	−1.266	27	0.216	0.288
TP9–300–320 ms	fake-real	2.547	27	0.017	**0.037**
	fake-no	0.881	27	0.386	0.421
	real-no	−1.835	27	0.077	0.132
FT10–300–320 ms	fake-real	2.327	27	0.072	0.132
	fake-no	0.744	27	0.820	0.820
	real-no	-0.060	27	0.040	0.080
TP10–300–320 ms	fake-real	1.655	27	0.110	0.176
	fake-no	0.375	27	0.711	0.742
	real-no	−1.490	27	0.148	0.222
FT9–500–520 ms	fake-real	3.580	27	0.001	**0.004**
	fake-no	-0.953	27	0.349	0.399
	real-no	−5.393	27	≤0.001	**0.004**
TP9–500–520 ms	fake-real	3.520	27	0.002	**0.007**
	fake-no	−1.077	27	0.291	0.349
	Real-no	−5.844	27	≤0.001	**0.004**
FT10–500–520 ms	fake-real	3.774	27	0.001	**0.004**
	fake-no	−1.160	27	0.256	0.323
	real-no	−6.525	27	≤0.001	**0.004**
TP10–500–520 ms	fake-real	3.303	27	0.003	**0.009**
	fake-no	−1.313	27	0.200	0.282
	real-no	−5.810	27	≤0.001	**0.004**

## Data Availability

Data available on request.

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
