# Peer review of "Brain Activities Show There Is Nothing Like a Real Friend in Contrast to Influencers and Other Celebrities"

_brainsci, 2023, doi:10.3390/brainsci13050831_

Round 1

Reviewer 1 Report

In this manuscript authors present brain activities regarding a real friendship and compare them with the activities regarding the influencers and other celebrities. After performed EEG recording, they derived the well known and expected conclusion that “there is nothing like a real friend”.

Topic of this manuscript is far, very, very far from areas of my expertise. However, regardless this fact, I cannot recognize contribution of the manuscript. In my opinion, authors was proving already proven things. Namely, authors treat well known and long time discussed topic of a real friendship. OK, authors compared a real friendship with influencers and other celebrities, but this topic have already considered and analyzed (with the same or very similar conclusions) in many humanity sciences, such as sociology, psychiatry, ... The conclusions performed based on the EEG recording is the only contribution I can see in the manuscript.

As a minor flaw, the manuscript contains some serious technical failings. For example, names of Section 3. and Sub-section 3.1 are on page 3, but main text of this Section and Subsection begins on the following page. In addition, Tables 1 and 2 technically seem poor (moreover, their ends are unreadable).

Author Response

Many thanks to reviewer 1. Respective comments and suggestions helped as a lot to improve our work. In the following, please find point-by-point responses:

Reviewer 1

General reviewer comment (copied from actual review): “In this manuscript authors present brain activities regarding a real friendship and compare them with the activities regarding the influencers and other celebrities. After performed EEG recording, they derived the well known and expected conclusion that “there is nothing like a real friend”.

Separate points raised by reviewer:

  • Reviewer comment: “Topic of this manuscript is far, very, very far from areas of my expertise. However, regardless this fact, I cannot recognize contribution of the manuscript. In my opinion, authors was proving already proven things. Namely, authors treat well known and long time discussed topic of a real friendship. OK, authors compared a real friendship with influencers and other celebrities, but this topic have already considered and analyzed (with the same or very similar conclusions) in many humanity sciences, such as sociology, psychiatry, ... The conclusions performed based on the EEG recording is the only contribution I can see in the manuscript”.

Our response: As reviewer 1 pointed out, crucially, real friendship was compared with influencer-based and celebrity-based friendship while both types of friendship were also contrasted with lack of friendship. It is indeed not the first time these phenomena are compared with each other, but most importantly, what reviewer 1 mentions as the only contribution of our work to the scientific literature actually is a novel contribution. To the best of our knowledge, no one has ever used the event-related brain potential approach like we did. Thus, while we understand that reviewer 1 finds our study not that innovative, which might be true in terms of theoretical concepts, but the inlcusion of event-related brain potentials in fact is novel.

  • Reviewer comment: “As a minor flaw, the manuscript contains some serious technical failings. For example, names of Section 3. and Sub-section 3.1 are on page 3, but main text of this Section and Subsection begins on the following page. In addition, Tables 1 and 2 technically seem poor (moreover, their ends are unreadable).

Our response: If we understand correctly, reviewer 1 refers to layout issues. If this is so, we can confirm that these are fixed now in the revised version of our manuscript. We agree that tables 1 and 2 needed to be refurbished. We of course followed this suggestion of reviewer 1.

Please, find the revised version of our manuscript attached to this response letter.

Thanks again!

Reviewer 2 Report

Authors present a study on 28 young participants who underwent EEG which was calculated into ERP (event related potentials) while getting exposed to pictures of "real" friends, "no friends", and "fake" friends, i.e. social media influencers who were followed by the participants in the digital world. Short (100ms) left frontal brain activity starting at around 250ms post stimulus was found to to process real friend and no friend names similarly, while both ERPs differed from those elicited by fake friend names;  real friend names elicited highest levels of brain activation. This is an interesting study and the manuscript is well written. 

1-It would be interesting to hear opinion of the authors on the fact that no-friends and real-friends elicited similar responses at FT9 and that no-friends and fake-friends elicit similar response at TP9. Maybe it would have been more useful to reduce the number of "real friends" to 2-3, which is probably a realistic number of friends in life - if even that.

2-In my opinion, this study has shown high brain activity and response in exposure to "people the participants knows personally" and I would not title this as a "friend". Even in the digital era, 20 "friends" is a lot. This might be only an issue of terminology, and a group of people of an average age of 21 years might think that a person can have 20 friends, but life experience shows that even at this age, even at groups of 20-30 people (sport clubs, chors, culture bands) there are subgroups in which people feel more comfortable. One idea for the future studies would be to compare 2-3 "best friends" setting to "real but in reality not real friends from social networks and life" to other groups. I suggest to include these aspects into limitations. Novelty of this study is that it attempts to show objectively differences in brain activity and does not relly on self-reporting. 

3-Please include future advances and possible clinical and sociobehavioral applications of this study. 

Author Response

Many thanks to reviewer 2 for all helpful comments and suggestions. In the following, please find point-by-point responses:

Reviewer 2

General reviewer comment (copied from actual review): “Authors present a study on 28 young participants who underwent EEG which was calculated into ERP (event related potentials) while getting exposed to pictures of "real" friends, "no friends", and "fake" friends, i.e. social media influencers who were followed by the participants in the digital world. Short (100ms) left frontal brain activity starting at around 250ms post stimulus was found to to process real friend and no friend names similarly, while both ERPs differed from those elicited by fake friend names;  real friend names elicited highest levels of brain activation. This is an interesting study and the manuscript is well written”.

              Our response: We want to thank reviewer 2 for this positive feedback. In addition, we appreciate the constructive comments and suggestions that helped us to improve our paper.

Separate points raised by reviewer:

  • It would be interesting to hear opinion of the authors on the fact that no-friends and real-friends elicited similar responses at FT9 and that no-friends and fake-friends elicit similar response at TP9. Maybe it would have been more useful to reduce the number of "real friends" to 2-3, which is probably a realistic number of friends in life - if even that.

Our response: Here, reviewer 2 points out one of the truly interesting findings of our study. As we mention in our discussion section, the left frontal (fronto-temporal area) includes the well-known Broca area, which among other mainly speech-motoric functions also processes holistic meaning that is expected to be different between influencers and other celebrities versus real friends (and relatives) and unknown people that at least have “normal” names (just like real friends and relatives). Further, we agree that actual deep friendship is rather limited, but it lies in the nature of an event-related potential study that one has to use at least 20 stimuli of a kind to ensure enough brain potential data to analyse. This is due to the averaging process that is meant to result in acceptable signal-to-noise ratios. In addition, we want to emphasise that for the condition of “real friends” we asked for names of “loved” friends, but also of “loved” relatives that our participants felt very close to.

We hope that this response satisfies reviewer 2.   

  • In my opinion, this study has shown high brain activity and response in exposure to "people the participants knows personally" and I would not title this as a "friend". Even in the digital era, 20 "friends" is a lot. This might be only an issue of terminology, and a group of people of an average age of 21 years might think that a person can have 20 friends, but life experience shows that even at this age, even at groups of 20-30 people (sport clubs, chors, culture bands) there are subgroups in which people feel more comfortable. One idea for the future studies would be to compare 2-3 "best friends" setting to "real but in reality not real friends from social networks and life" to other groups. I suggest to include these aspects into limitations. Novelty of this study is that it attempts to show objectively differences in brain activity and does not rely on self-reporting. 

Our response: This point is similar to point 1. However, here reviewer 2 suggests changing terminology, because he or she repeats that having 20 “real” friends is not realistic. As pointed out in our response to point 1, we asked our participants for names of “loved” friends and relatives, which we now understand is a mix up of two distinguishable groups. We agree that this should be dealt with in the discussion section as a potential limitation. Thus, we added this point in the discussion section in the revised version of our manuscript.

Here is what we added:

“However, we would like to mention that the condition “real friends”, which includes actual loved friends as well as loved relatives could be understood as representing close loved people in general and not particularly friends. It might seem unlikely, especially for an older generation, that one can have 20 close friends. Consequently, the condition “real friends” might be seen as a limitation in terms of terminology”.

  • Please include future advances and possible clinical and sociobehavioral applications of this study.

Our response: We did add a short paragraph at the end of the discussion section.

Thanks again for your highly appreciated efforts!

Please, find the revised version of our manuscript attached to the response letter.

Best regards!

Reviewer 3 Report

The authors proposed a study about brain activation of a group of people in three different conditions. The participants provided a personal list of their favourite celebrities (fake friend names), real friends or close relatives (real friend names) and names of people to whom they feel no closeness (no friend names). The selected names were shown to them in a random sequence and their brain activities were recorded by electroencephalography. ERPs elicited by each condition were analyzed and compared. The results showed a similar levels of brain activation for real friend names and no friend names in the left fronto-temporal region starting at around 250 ms post stimulus onset. Moreover, real friend names and no friend names elicited higher cortical brain activation compared to fake friends name. A second effect was found starting at around 300ms post stimulus onset in left and right frontal and temporo-parietal regions. It revealed that real friend names elicited higher brain activities compared to both other conditions (fake friend names and no friend names).

This study could be a starting point for further investigations but it needs to be improved. I have some doubts and suggestions. 

1) In my opinion, at this stage of your study there is not strong evidence to draw the conclusions you have described. They should be rewritten highlighting that yours are more hypotheses than certainty.

2) You refer to the topic of social media addiction, but your study does not consider the correlation between the use of social media and brain activity. The participants were divided into three groups according to time spent on social media, so an analysis of brain activity for each group could provide more solidity to the results. 

3) Add the limitations of this study and its future developments.

4) How long are the epochs considered (line 133)?

5) Add a legend for the colours in Figure 1.

6) Fix the formatting of the tables.

Author Response

Many thanks to reviewer 3 for all helpful comments and suggestions. In the following please find point-by-point responses:

Reviewer 3

General reviewer comment (copied from actual review): “The authors proposed a study about brain activation of a group of people in three different conditions. The participants provided a personal list of their favourite celebrities (fake friend names), real friends or close relatives (real friend names) and names of people to whom they feel no closeness (no friend names). The selected names were shown to them in a random sequence and their brain activities were recorded by electroencephalography. ERPs elicited by each condition were analyzed and compared. The results showed a similar levels of brain activation for real friend names and no friend names in the left fronto-temporal region starting at around 250 ms post stimulus onset. Moreover, real friend names and no friend names elicited higher cortical brain activation compared to fake friends name. A second effect was found starting at around 300ms post stimulus onset in left and right frontal and temporo-parietal regions. It revealed that real friend names elicited higher brain activities compared to both other conditions (fake friend names and no friend names). This study could be a starting point for further investigations but it needs to be improved. I have some doubts and suggestions”.

Our response: We are very happy about the positive feedback of reviewer 3 and we highly appreciate his or her comments that helped us to improve our manuscript. We are particularly happy about the last comment that our study could be a starting point for future investigations after thorough revision, which we hope we achieved.

Separate points raised by reviewer:

  • In my opinion, at this stage of your study there is not strong evidence to draw the conclusions you have described. They should be rewritten highlighting that yours are more hypotheses than certainty.

Our response: We must admit that our study is not really hypotheses driven, it is quite explorative, which means that its result can be seen as leading to hypotheses rather than answering questions. We agree that this should be highlighted.

We added the following in the discussion section:

“However, we would like to mention that the condition “real friends”, which includes actual loved friends as well as loved relatives could be understood as representing close loved people in general and not particularly friends. It might seem unlikely, es-pecially for an older generation, that one can have 20 close friends. Consequently, the condition “real friends” might be seen as a limitation in terms of terminology. In addi-tion, due to the innovative nature of this study in terms of using ERPs to investigate social media impact, or more accurately said, brain processes related to people’s names varying in real versus fake closeness, our findings should more be seen as leading to hypotheses rather than answering questions. This study is more explorative than driven by hypotheses and thus can be a starting point for future investigations”.

  • You refer to the topic of social media addiction, but your study does not consider the correlation between the use of social media and brain activity. The participants were divided into three groups according to time spent on social media, so an analysis of brain activity for each group could provide more solidity to the results.

Our response: In our experiment, we did not intend to cover the topic of social media addiction. However, we asked our participants about their average daily use of social media as a demographic aspect. With this we only wanted to make sure that all participants are actually social media users. 23 participants (out of 28) reported to stay below 3 hours per day, 16 even below 2 hours, which means that our participants were not addicted. Only 5 participants reported usage of more than 3 hours. It would indeed be very interesting to build groups and calculate new ERPs, but this would be an own study. We like the idea and maybe we reanalyse our data for another publication, but it would be too much for this paper. But, we think it would be good to mention under “participants” that the respective data were not further analysed, because they simply are meant to demonstrate that our participants are social media users.

  • Add the limitations of this study and its future developments.

Our response: These points were also mentioned by another reviewer. Two main limitations have already been added, one being the point that the condition “real friends” is based on names of loved real friends, but also loved close relatives. This widens the term “friendship” a bit, but it is only a terminology issue, if at all. What is common to both groups is positive, affective closeness to people out of real life. We added this limitation. The second limitation is defined as what reviewer 3 mentioned in point 1. The study is more explorative than driven by hypotheses. Thus, it might lead to new hypotheses rather than answer questions. We also added this point as a limitation.

  • How long are the epochs considered (line 133)?

Our response: All epochs are 1.1s long. Please, see the respective information in our method section:

“An offline bandpass filter from 0.1 to 30Hz was applied before generating epochs from 100ms before stimulus onset (baseline) until 1s after stimulus onset. All epochs contaminated by visible artifacts were manually selected and excluded, and those with the electrooculogram (EOG) amplitude exceeding ± 75mV were automatically excluded”.

  • Add a legend for the colours in Figure 1.

Our response: Good point, thanks heaps! Done.

  • Fix the formatting of the tables.

Our response: Yes, that needed to be done as we realised ourselves. It must have been a strange formatting issue. Thanks for mentioning this. Done!

Thanks again and best regards!

Round 2

Reviewer 1 Report

My main demand from the first review round remains. Namely, in the manuscript, authors are proving already proven things. Of course, the provement is performed on an innovative manner (the event-related brain potential approach is used for the first time, as authors said/declared in their response), but remain that the already proven and well considered topic of friendship (real friendship vs influencer-based and other celebrities-based friendship) is observed, that authors also confirmed (as they said, which might be true in terms of theoretical concepts) in their response. Therefore, in my opinion, authors should firstly theoretically presented the manuscript topic consideration in humanity sciences and then justify why they use the event-related brain potential approach in order to prove already proven and considered topic. In particular, why it is important to consider friendship topic in this way, when the similar conclusions are derived in the other ways. In my opinion, authors must first clarified these things and after that the manuscript should be resubmitted.

Authors have satisfied my minor demands from the first review round, but a new technical failing, similar as in the original manuscript version, is appeared. Namely, name of Section 2 is in the end of page 2, but main text of this Section begins on the following page. Section 2 name should be placed on the same page where main text of this section begins.

Author Response

Dear reviewer 1, thanks for your helpful comments and suggestions. In the second revision round we included the following paragraph in the introduction section:

"Although the definition of friendship has been found difficult by a number of sociologists in the past [e.g. 8,9], there clearly are crucial differences compared to what influencers, bloggers and youtubers mean to us. For instance, Blau [10] investigated the topic of friendship by asking questions to people of varying generations and analyzing their responses. The result has been an understanding of friendship being restricted to a very limited set of close friends. Allan [11] summed up three important points for a friendship understood as a personal relationship. Point 1 is that it must be a relationship between individuals, point 2 is that it must be a private relationship and point 3 is that it must involve the person as the person he or she really is.

 From a neuroethological perspective, friendship is defined through the quality of interactions between individuals. Two humans are friends when they engage in a bidirectional and affiliative way with a higher frequency than they do it with nonfriends. Respective interactions should be nonreproductive and consistent over time [12,13]. According to Dunbar [14], friends share their lives in a way different to just casually meeting strangers including emotional bonds and regular contact.

Whereas a group relationship has been understood as between mutually substitutable people, friendship is a relationship with someone not replaceable [15]. Crucially, a very important aspect of friendship as mentioned in various reports is its non-exploitive nature [e.g. 11]. Joy has to be the driving motive, not some instrumental reason [11]. This latter point might be most important in the context of influencers, bloggers and youtubers.

........and a bit further down:

"Even though the topic of friendship has been researched a lot in the past (see above), what has been neglected so far is the use of brain imaging technology, especially in the context of the fresh and acute problem of fake friendship of influencers and other celebrities".

We hope that this satisfies your point.

We are very thankful for this suggestions, because we feel it helped us a lot to improve our manuscript.

The second point about "technical failing" is simply a layout formatting issue. Once, the paper is formatted by the journal editors those issues are sorted out. However, it should already be okay in the newly revised version of our manuscript.

Thanks again and with very best wishes!

Reviewer 3 Report

The authors addressed my requests and the paper improved.

Author Response

Dear reviewer 3, thanks you so much again for helping us to improve our contribution.

Best regards!